

# Comparative analyses of the gut microbiome of two sympatric rodent species, *Myodes rufocanus* and *Apodemus peninsulae*, in northeast China based on metagenome sequencing

Jing Cao[1], Shengze Wang[2], Ruobing Ding[1], Yijia Liu[1] and Baodong Yuan[2]

[1] College of Biology and Food, Shangqiu Normal University, Shangqiu, Henan, China
[2] School of Life Science, Liaocheng University, Liaocheng, Shandong, China

## ABSTRACT

The gut microbiota is integral to an animal's physiology, influencing nutritional metabolism, immune function, and environmental adaptation. Despite the significance of gut microbiota in wild rodents, the Korean field mouse (*Apodemus peninsulae*) and the gray red-backed vole (*Myodes rufocanus*) remain understudied. To address this, a metagenomic sequencing analysis of the gut microbiome of these sympatric rodents in northeast China's temperate forests was conducted. Intestinal contents were collected from *A. peninsulae* and *M. rufocanus* within the Mudanfeng National Nature Reserve. High-throughput sequencing elucidated the gut microbiome's composition, diversity, and functional pathways. Firmicutes, Bacteroidetes, and Proteobacteria were identified as the dominant phyla, with *M. rufocanus* showing greater microbiome diversity. Key findings indicated distinct gut bacterial communities between the species, with *M. rufocanus* having a higher abundance of Proteobacteria. The gut microbiota of *A. peninsulae* and *M. rufocanus* differed marginally in functional profiles, specifically in the breakdown of complex carbohydrates, which might reflect their distinct food preferences albeit both being herbivores with a substantial dietary overlap. The investigation further elucidated gut microbiota's contributions to energy metabolism and environmental adaptation mechanisms. This study aligns with information on rodent gut microbiota in literature and highlights the two understudied rodent species, providing comparative data for future studies investigating the role of gut microbiota in wildlife health and ecosystem functioning.

## INTRODUCTION

The gut microbiota in animals comprises a complex community of symbiotic microorganisms, which is essential for the host's digestive system (*Lynch & Hsiao, 2019*). This microbiota plays a pivotal role in various aspects of the host's physiology, including nutritional metabolism (*Bäckhed et al., 2004*; *Turnbaugh et al., 2006*), immune function (*Cani et al., 2007*; *Vijay-Kumar et al., 2010*), environmental adaptation (*Ma et al., 2019*;

Corresponding author
Baodong Yuan,
yuanbao365@163.com

*Zhang et al., 2024*; *Zhu et al., 2024*), and behavior (*Parashar & Udayabanu, 2016*; *Sharon et al., 2019*). The gut microbiota demonstrates limited stability and is subject to modulation by various factors, including environmental factors (*Koziol et al., 2023*; *Risely et al., 2023*), maternal delivery (*Collado et al., 2008*), genetics (*Bäckhed et al., 2004*; *Turnbaugh et al., 2006*), geography (*Wang et al., 2022*), lifestyle (*Risely et al., 2022*; *Víquez-R et al., 2021*) and human encroachment (*Fackelmann et al., 2021*; *Heni et al., 2023*). Among these, diet is one of the most significant determinants of gut microbiota diversity. Herbivores, with their reliance on plant-based materials, host diverse microbial communities capable of breaking down complex polysaccharides, whereas carnivores exhibit reduced microbial diversity (*de Jonge et al., 2022*). Whole metagenomic sequencing has revealed that the Amur tiger (*Panthera tigris altaica*) possesses a typical carnivorous intestinal microbial structure, and its main functions are carbohydrate metabolism, amino acid metabolism, and membrane transport (*He et al., 2018*). In contrast to carnivores, the giant panda (*Ailuropoda melanoleuca*) and the red panda (*Ailurus fulgens*) demonstrates a higher abundance of bacteria that are proficient in digesting cellulose and hemicellulose, thereby facilitating the digestion of bamboo (*Kong et al., 2014*; *Li et al., 2015*; *Zhu et al., 2011*).

Rodents are one of the most diverse and widely distributed orders within the class Mammalia. They inhabit various ecological niches, ranging from herbivorous granivores to opportunistic omnivores. This significant diversity renders rodents an exemplary model for investigating the functional and evolutionary dynamics of gut microbiomes. The gut microbiome of rodents is predominantly composed of bacterial phyla, including Firmicutes, Bacteroidetes, Proteobacteria, and Actinobacteria. Notable genera, such as *Bacteroides*, *Lactobacillus*, and *Prevotella*, are frequently linked to the dietary niches occupied by rodents (*Wang et al., 2022*). The gut microbiota plays a crucial role in energy extraction, immune system modulation, and detoxification processes, thereby facilitating the adaptation of rodents to diverse diets and habitats (*de Jonge et al., 2022*). Rodent microbiomes reflect evolutionary pressures, with herbivorous species harboring bacteria specialized in fiber digestion (*de Jonge et al., 2022*). The bank voles cecal microbial community includes dominant bacterial phyla such as Firmicutes and Bacteroidetes, which comprises key genera like *Ruminococcus* and *Treponema*, associated with fiber fermentation (*Kohl et al., 2016*). Woodrat gut microbiota consists of bacterial genera, such as *Coprococcus*, *Lactobacillus*, *Oxalobacter*, *Bacillus*, *Enterococcus*, and *Clostridium* capable of degrading various PSCs (*Kohl & Dearing, 2016*). The gut microbiome of the capybara, the largest living rodent, is highly specialized for efficiently degrading plant-derived lignocellulosic biomass. Specifically, Fibrobacteres primarily degrade cellulose through adhesion and enzymatic mechanisms, while Bacteroidetes are specialize in breaking down hemicellulose and pectin *via* polysaccharide utilization-associated loci and carbohydrate-active enzyme clusters (*Cabral et al., 2022*). In addition, the dynamic changes within the intestinal microbiome represent a crucial adaptive mechanism for rodents in response to environmental conditions. Significant seasonal variations in the gut microbiota of wild wood mice (*Apodemus sylvaticus*) are likely due to dietary shifts from insects to seeds (*Maurice et al., 2015*). Research on free-ranging woodrat (*Neotoma* spp.) has shown that seasonal changes in the diet lead to the restructuring of gut microbial communities,

facilitating compositional convergence and adaptation to the host's changing diet (*Klure & Dearing, 2023*).

As keystone species within forest ecosystems, the Korean field mouse (*Apodemus peninsulae*) and gray red-backed vole (*Myodes rufocanus*) serve not only as primary consumers and seed dispersers but are also a significant food source for carnivores. However, the structural and functional attributes of the intestinal microbiota in *A. peninsulae* and *M. rufocanus* remain unexplored in their natural habitats. *A. peninsulae* is classified under the *Apodemus* family, whereas *M. rufocanus* belongs to the *Rattus* family within the Glires order. They are among the dominant species within the temperate forests of northeastern China. Their ecological niches exhibit some overlap as they inhabit various environments, including forests, shrublands, glades, grasslands, and the peripheries of agricultural areas. *A. peninsulae* predominantly consumes roots, grains, seeds, berries, and nuts (*Batsaikhan et al., 2016*), whereas *M. rufocanus* mainly feeds on the vegetative parts of grasses, herbs, and dwarf shrubs in addition to berries (*Henttonen & Viitala, 1982*; *Sulkava, 1999*).

Considering the substantial overlap in the ecological niches and diets between *A. peninsulae* and *M. rufocanus*, we hypothesize that their gut microbiota exhibit marked convergence in taxonomic composition and functional characteristics, with similarities surpassing those attributable to host evolutionary divergence. To test such a hypothesis, this study collected intestinal contents from *A. peninsulae* and *M. rufocanus* at the Mudanfeng National Nature Reserve in October 2022. The gut microbiome's composition and diversity of the two species were analyzed and compared by high-throughput sequencing. Finally, this study inferred the functional pathways of the gut microbiome. This study provides a baseline for understanding the gut microbial community and potential functions for two rodent species coexisting in Chinese forests.

## MATERIALS AND METHODS

### Ethics statement

The entire study procedure adhered to the natural wildlife protection law and did not produce any substances harmful to the environment and animals. All experimental procedures were conducted in accordance with the guidelines established by the Animal Experiment Ethics Committee of the Shangqiu Normal University (Approval number 2022102). The mice were housed in a temperature-controlled room under a 12-h light/dark cycle. The intestinal contents were sampled post-euthanasia, using an overdose of sodium pentobarbitone anesthesia to ensure that the animals did not experience any pain or distress. Potential post-operative pain was mitigated by administering analgesics prophylactically. The mice were closely monitored throughout the study, and any signs of distress or discomfort were addressed immediately.

### Sample collection

All animals were trapped using Sherman live cages measuring $28 \times 15 \times 15$ cm within the mixed coniferous and broadleaf forest of the Mudanfeng National Nature Reserve, located in the northeastern region of China, in October 2022. Peanut butter was used as bait and
placed at the far end, past the trigger plate. The traps were opened after 24 h; all mice were alive when the traps were opened. Seventeen *A. peninsulae* and 18 *M. rufocanus* were captured in total. After being taken back to the laboratory, the animals were allowed to stand still for 30 min. Subsequently, they were euthanized by cervical dislocation under aseptic conditions. After the disinfection of the abdominal surface with alcohol-soaked cotton, the animals were dissected, and the intestines were promptly extracted and immersed in a mortar containing liquid nitrogen. The intestinal contents were transferred into a marked, sterile freezing tube and stored at −80 °C.

## DNA extraction and metagenomic sequencing

Microbial DNA was extracted from the frozen intestinal contents using the QIAamp® Fast DNA Stool Mini Kit (Qiagen, Hilden, Germany) according to the manufacturer's protocols. The TIANquick Midi purification kit (Tiangen, Beijing, China) was used for DNA fragment purification after the terminal end repair. An A base was added to the 3′-end of the purified DNA fragments, and a sequencing adaptor was added to the 3′ and 5′-ends. The DNA purity and concentration were determined using agarose gel electrophoresis (1%) (*Porebski, Bailey & Baum, 1997*). A total of six DNA samples from *A. peninsulae* (three females and three males) and six from *M. rufocanus* (three females and three males), allof sufficient quality, were utilized. Detail information on the wild rodents captured and used in this study is included in Table 1. The DNA library was constructed following the TruSeq DNA Sample Preparation Guide (15026486 Rev.C; Illumina). The library quantities were assessed using a Qubit™ 2.0 fluorometer (Thermo Fisher Scientific, Waltham, MA, USA) and a quantitative polymerase chain reaction method. Finally, the libraries were sequenced using an Illumina PE 150 platform (Illumina, San Diego, CA, USA) to produce 150 bp paired-end reads.

## Quality control and assembly of the sequencing data

Sequencing data were aligned to the mouse reference genome using the Bowtie2 version 2.5 (*Langmead & Salzberg, 2012*), and reads aligning to the reference sequence were removed to ensure no contamination of the host DNA in the sequencing data. Subsequently, sequences that were not aligned to the host  were trimmed using Trimmomatic software version 0.4 (*Bolger, Lohse & Usadel, 2014*): (1) adapter sequences with overlaps exceeding 5 bp; (2) sequences containing >10% N nucleotides, and (3) low-quality bases with a Phred quality score (*Q*-value) of ≤10, which accounted for >50% of the total bases. After filtering, the resulting clean data from all samples were merged, spliced, and assembled using the MEGAHIT software (*Liu et al., 2015*), with default settings employing the de Bruijn graph principle. Based on the overlaps among k-mers, a de Bruijn graph was constructed, and contigs were generated. Contigs exceeding 800 bp were used for subsequent analyses.

## Operational taxonomic unit clustering and species annotation

The open reading frame prediction of the spliced contig was conducted using Prodigal software (*Hyatt et al., 2010*), and the redundant results were eliminated by CD-HIT software (*Fu et al., 2012*) to obtain the initial gene catalog. Subsequently, a 95% identity and 90% coverage threshold were applied for clustering, with the longest sequence designated

**Table 1  Statistical information of sample data.**

| Sample name | Sex | Total reads | Clean reads | Percentage | GC content | %>Q20 | % >Q30 | Average length | N50 |
|---|---|---|---|---|---|---|---|---|---|
| Mruf1 | Male | 38,743,870 | 38,368,224 | 99.03% | 43.42% | 98.05% | 94.24% | 680.4 | 649 |
| Mruf2 | Male | 41,646,142 | 41,268,366 | 99.09% | 43.66% | 98.01% | 94.18% | 686.9 | 655 |
| Mruf3 | Male | 46,773,472 | 46,328,760 | 99.05% | 43.51% | 98.12% | 94.46% | 704.8 | 677 |
| Mruf4 | Female | 37,960,856 | 37,573,294 | 98.98% | 45.72% | 98.03% | 94.30% | 768.4 | 691 |
| Mruf5 | Female | 45,416,240 | 44,935,158 | 98.94% | 45.84% | 98.12% | 94.50% | 777.5 | 709 |
| Mruf6 | Female | 46,124,394 | 45,592,170 | 98.85% | 45.97% | 98.07% | 94.37% | 778.2 | 718 |
| Apen1 | Male | 45,926,698 | 45,334,568 | 98.71% | 45.63% | 97.87% | 93.89% | 1,065.10 | 1,083 |
| Apen2 | Male | 46,679,116 | 46,054,064 | 98.66% | 45.63% | 97.70% | 93.49% | 1,057.50 | 1,066 |
| Apen3 | Male | 47,984,784 | 47,430,632 | 98.85% | 45.15% | 98.01% | 94.28% | 1,057 | 1,054 |
| Apen4 | Female | 38,723,404 | 38,350,526 | 99.04% | 45.92% | 97.77% | 93.57% | 771 | 691 |
| Apen5 | Female | 39,804,916 | 39,410,234 | 99.01% | 45.79% | 97.87% | 93.82% | 770.4 | 690 |
| Apen6 | Female | 44,963,386 | 44,517,644 | 99.01% | 46.21% | 98.09% | 94.33% | 766.1 | 688 |

the representative sequence. Clean reads from each sample were aligned with the gene catalog (at 95% identity) using bowtie2 software (*Langmead & Salzberg, 2012*), and gene abundance data for each sample were tabulated. The DIAMOND software (*Buchfink, Xie & Huson, 2015*) was used to align the unigenes against the National Center for Biotechnology Information NR database of bacteria using BLASTP with a cutoff value of 1e−5. Hits with $e$-values $\leq$ 10 times the minimum $e$-value were chosen for further analysis. MEGAN software was employed to annotate the species of the sequences using the lowest common ancestor algorithm (*Huson et al., 2018*).

### Diversity analysis

Principal component analysis (PCA) was conducted using R to delineate the distances between samples based on the compositional similarity of different samples at the 97% threshold (*Langfelder & Horvath, 2008*). Analysis of similarities (Anosim) was employed to determine if the differences between the groups were significantly larger than those within the groups. $\alpha$-Diversity, assessed using the Chao1, Ace, Shannon, and Simpson indices, was analyzed using R to evaluate differences in gut bacterial diversity between the two species. The Chao1 and Ace indices measure species richness, expressed as species count. The Shannon and Simpson indices were utilized to estimate species diversity. STAMP analysis was conducted at the phylum, family, and genus levels to compare species distribution and relative abundance between the two groups (*Parks et al., 2014*).

### Function prediction

Utilizing DIAMOND software (*Buchfink, Xie & Huson, 2015*), the gene catalog was aligned against the Kyoto Encyclopedia of Genes and Genomes (KEGG) (*Kanehisa et al., 2016*) and the Carbohydrate Enzyme (CAZy) databases (*Drula et al., 2022*) to identify the most similar functional annotations using a cutoff of 1e−5. For every sequence, the alignment possessing the highest score, identified by one high-scoring pair exceeding 60 bits, was chosen for further analysis. The quantities and relative abundances of genes across various

functional levels were determined. Cluster analysis was conducted using PCA for dimension reduction. Through linear discriminant analysis effect size (LEfSe), multivariate statistical analysis, and STAMP analysis of metabolic pathways, differences in the functional profiles of the intestinal microbiota between groups were investigated.

## RESULTS

### Sequencing of the gut microbiome

The statistical information of the sample data is shown in Table 1. A total of 520,747,278 reads were obtained from the 12 samples. After quality control, 515,163,640 clean reads were derived. An average of 42,930,303 sequences were generated per sample (Table 1). The screened sequences had an average GC content of 45.20%, and the average base ratios for bases with mass values exceeding 20 and 30 were 97.98% and 94.12%, respectively. The rarefaction curve (Fig. S1) demonstrated that once the sequencing data reaches 20,000 reads, the sample curve plateaus, indicating that the sequencing depth sufficiently captures the microbiome diversity, thus validating the reliability of the subsequent analysis.

### Gut bacterial composition

A total of 43 phyla, 108 classes, 227 orders, 549 families, 1,761 genera, and 6,406 species were identified in the two species. Dominant at the phylum level in *A. peninsulae* were Bacteroidetes (33.14% ± 9.38%), Firmicutes (31.04% ± 2.87%), Proteobacteria (19.69% ± 0.76%), and Actinobacteria (13.73% ± 5.11%), comprising ~97.6% of the bacterial community (Fig. 1A). In *M. rufocanus*, the dominant phyla included Proteobacteria (29.38% ± 0.86%), Bacteroidetes (26.38% ± 5.24%), Firmicutes (23.77% ± 5.15%), and Actinobacteria (12.37% ± 2.75%), collectively constituting ~91.83% of the bacterial community.

At the family level in *A. peninsulae*, the following were predominant: Muribaculaceae (21.52% ± 10.08%), Lactobacillaceae (9.19% ± 2.89%), Lachnospiraceae (6.58% ± 2.82%), Eggerthellaceae (5.67% ± 2.54%), Pasteurellaceae (3.71% ± 1.39%), Enterobacteriaceae (2.52% ± 1.5%), Bacteroidaceae (2.23% ± 0.77%), Bacillaceae (1.93% ± 0.32%), Rikenellaceae (1.93% ± 0.58%), and Streptomycetaceae (1.89% ± 0.06%). In *M. rufocanus*, the dominant families included Muribaculaceae (11.7% ± 4.04%), Lachnospiraceae (5.04% ± 1.37%), Clostridiaceae (4.64% ± 1.19%), Eggerthellaceae (4.57% ± 2.07%), Desulfovibrionaceae (4.08% ± 1.05%), Nostocaceae (4.0% ± 1.4%), Bacillaceae (2.63% ± 0.68%), Prevotellaceae (2.58% ± 0.23%), Streptomycetaceae (2.39% ± 0.17%), and Flavobacteriaceae (2.26% ± 0.29%; Fig. S2).

At the genus level (Fig. 1B), the gut bacteria of *A. peninsulae* were largely dominated by *Duncaniella* (26.4% ± 14.27%), *Adlercreutzia* (8.49% ± 4.98%), *Pasteurella* (7.17% ± 2.29%), *Bacteroides* (4.91% ± 2.41%), *Lachnoclostridium* (4.57% ± 2.46%), *Alistipes* (4.21% ± 1.88%), *Streptomyces* (3.76% ± 0.4%), *Clostridium* (2.82% ± 0.31%), *Bacillus* (2.62% ± 0.43%), and *Pseudomonas* (2.33% ± 0.22%) (Fig. 1B). In *M. rufocanus*, the top genera were *Duncaniella* (14.88% ± 4.51%), *Clostridium* (9.09% ± 3.17%), *Nostoc* (8.24% ± 3.62%), *Desulfovibrio* (7.92% ± 1.77%), *Muribaculum* (6.67% ± 2.22%), *Adlercreutzia* (5.08% ± 2.2%), *Prevotella* (4.87% ± 0.25%), *Streptomyces* (4.79% ± 0.09%), *Pseudomonas*

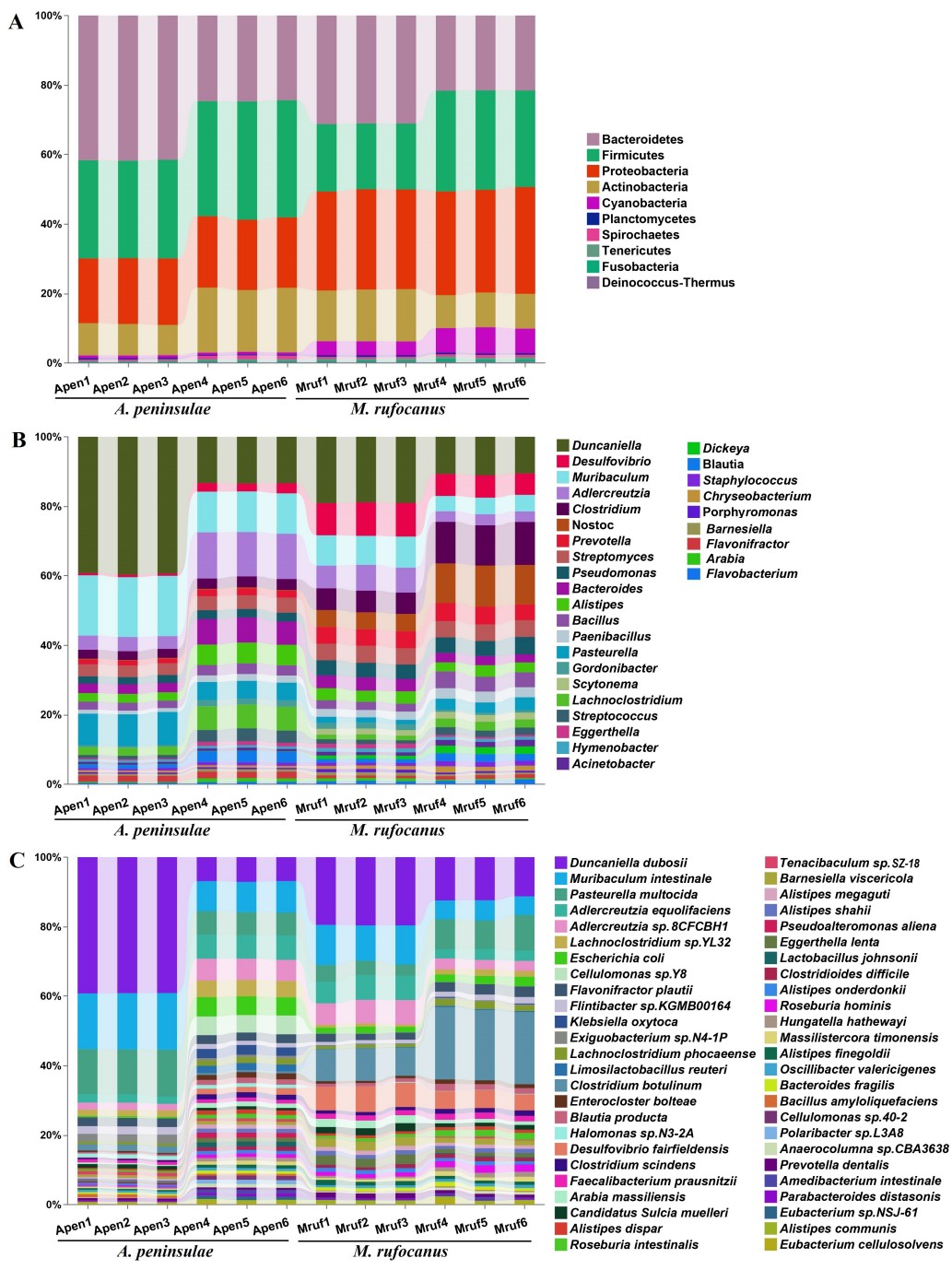

**Figure 1** **Taxonomic annotation of the gut community from *A. peninsulae* and *M. rufocanus*.** (A) Taxonomic annotation at the phylum level, depicting the top 10 phyla. (B) Taxonomic annotation at the genus level, featuring the top 30 genera. (C) Taxonomic annotation at the species level, featuring the top 50 species.

(4.4% ± 0.32%), and *Bacteroides* (3.14% ± 0.63%). Therefore, *Duncaniella*, *Adlercreutzia*, *Bacteroides*, *Streptomyces*, and *Clostridium* are the dominant bacterial genera shared by the gut community of the two species.

At the species level (Fig. 1C), the gut bacteria of *A. peninsulae* were largely dominated by *Duncaniella duboisii* (23.05% ± 17.65%), *Muribaculum intestinale* (12.52% ± 4.03%), *Pasteurella multocida* (9.84% ± 3.55%), *Adlercreutzia equolifaciens* (4.64% ± 2.51%), *Adlercreutzia* sp. 8CFCBH1 (4.08% ± 2.29%), *Lachnoclostridium* sp. YL32 (3.29% ± 1.60%), *Escherichia coli* (2.91% ± 2.73%), *Cellulomonas* sp. Y8 (2.64% ± 2.88%), *Flavonifractor plautii* (2.49% ± 0.07%), and *Flintibacter* sp. KGMB00164 (1.76% ± 0.53%). In *M. rufocanus*, the top 10 species were *D. duboisii* (15.81 ± 4.15%), *Clostridium botulinum* (14.78% ± 6.42%), *M. intestinale* (8.26% ± 3.05%), *P. multocida* (6.53% ± 2.86%), *Desulfovibrio fairfieldensis* (6.01% ± 1.18%), *A. equolifaciens* (4.87% ± 2.13%), *Adlercreutzia* sp. 8CFCBH1 (4.78% ± 2.11%), *F. plautii* (2.54% ± 0.49%), *E. coli* (2.08% ± 0.44%), and *Barnesiella viscericola* (2.01% ± 0.55%).

## Microbiome diversity analyses

A PCA-normalized normalized distribution plot highlighted the distinct differences in the bacterial communities between the two species. The first two principal components accounted for 77.13% of the variation, with PCA1 explaining 54.2% and PCA2 22.93%. A greater dispersion in *A. peninsulae* than *M. rufocanus* suggests a higher likelihood of intraspecific variability within *A. peninsulae* (Fig. 2A). Consequently, ANOSIM was employed to assess if significant differences existed between the two species. The results indicated statistically significant differences ($R = 0.844$, $P = 0.002$), suggesting distinct community structures between *A. peninsulae* and *M. rufocanus* (Fig. 2B).

$\alpha$-Diversity was evaluated to compare the diversity of gut bacteria between the two species. The Chao index for *A. peninsulae* exceeded that of *M. rufocanus* ($p > 0.05$), suggesting no significant difference in gut bacterial species richness between the species (Fig. 3). The Simpson (0.11 ± 0.02) and Shannon (2.29 ± 0.06) indices for *M. rufocanus* were higher than those for *A. peninsulae* (Simpson: 0.098 ± 0.005, Shannon: 2.1 ± 0.12), indicating greater abundance and evenness of gut bacterial species in *M. rufocanus*. Furthermore, the Shannon index revealed that the microbial community diversity within the gut of *M. rufocanus* was significantly greater than within *A. peninsulae*.

## Differences in the gut microbiome composition

Differentially abundant taxa between *A. peninsulae* and *M. rufocanus* gut samples were identified using LEfSe analysis, which screened for significantly distinct biomarkers between the two species. The key microbiomes of *A. peninsulae* and *M. rufocanus* exhibited distinct differences from one another (Fig. 4). Taxa unique to *A. peninsulae* include species *Ligilactobacillus animalis* and *Ligilactobacillus murinus* (Bacilli; Lactobacillales; Lactobacillaceae; *Ligilactobacillus*); *M. intestinale* (*Muribaculum*); and *P. multocida* (Pasteurellales; Pasteurellaceae; *Pasteurella*). In contrast, *C. botulinum* (Bacillota; Clostridia;

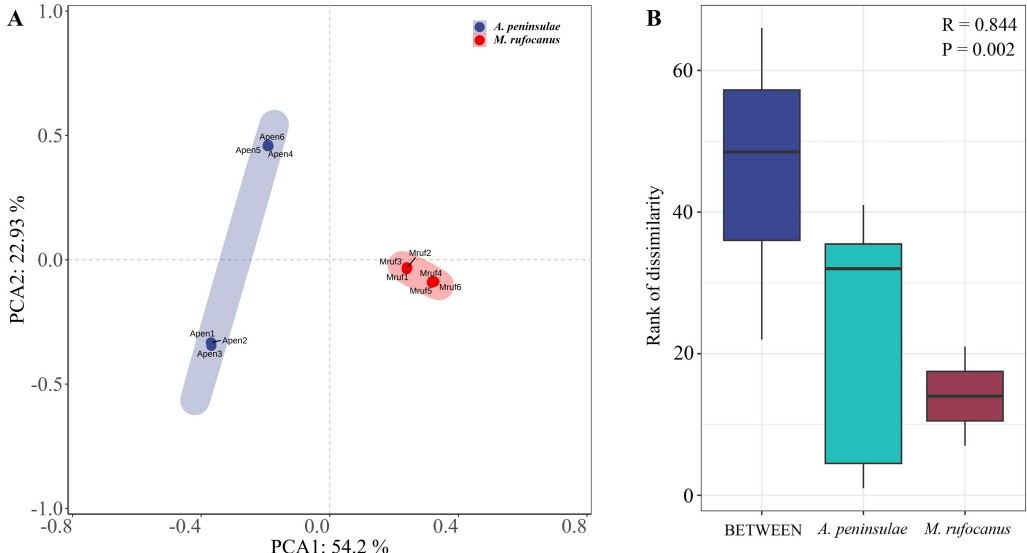

**Figure 2** **Differences in bacterial communities between *A. peninsulae* and *M. rufocanus*.** (A) PCA of structure differentiation and interindividual similarity in the gut microbiota. (B) Significant differences in the gut microbiota between the two rodent species as indicated by ANOSIM.

Eubacteriales; Clostridiaceae; *Clostridium*) and *Nostoc edaphicum* (Cyanobacteriota; Cyanophyceae; Nostocales; Nostocaceae; *Nostoc*) are specific to *M. rufocanus.*

This study screened for significantly distinct biomarkers between the different sexes (Fig. 5). Compared to female *A. peninsulae*, the key microbiomes of male individuals include *D. duboisii* (Bacteroidota; Bacteroidia; Bacteroidales; Muribaculaceae; *Duncaniella*), *L. murinus* (Bacillota; Bacilli; Lactobacillales; Lactobacillaceae; *Ligilactobacillus*), *M. intestinale* (Bacteroidota; Bacteroidia; Bacteroidales; Muribaculaceae; *Muribaculum*), *P. multocida* (Pseudomonadota; Gammaproteobacteria; Pasteurellales; Pasteurellaceae; *Pasteurella*), and *L. animalis* (Bacillati; Bacillota; Bacilli; Lactobacillales; Lactobacillaceae; *Ligilactobacillus*). The key microbiomes for female *A. peninsulae* are *Cellulomonas* sp. Y8 (Actinomycetota; Actinomycetes; Micrococcales; Cellulomonadaceae; *Cellulomonas*). In male *M. rufocanus*, the key microbiome is *D. duboisii.* In contrast, *Clostridium* (Bacteroidota; Bacteroidia; Bacteroidales; Muribaculaceae; *Duncaniella*) is the key genus in female *M. rufocanus.*

## Differences in the functional profiles of the gut microbiome

To delineate the functional composition of the gut microbiome and discover the functional differences between the two species, a functional profile analysis was performed using the KEGG and CAZy databases based on trimmed metagenomic data.

Metagenomic analysis confirmed 4,508 KEGG orthologous categories, comprising five at level A, 54 at level B, and 495 at level C. At level A, the gut microbiome of both species was predominantly characterized by metabolism, genetic information processing, and environmental information processing (Fig. 6A). Within the top 30 pathways at level B, the dominant categories included protein families associated with genetic

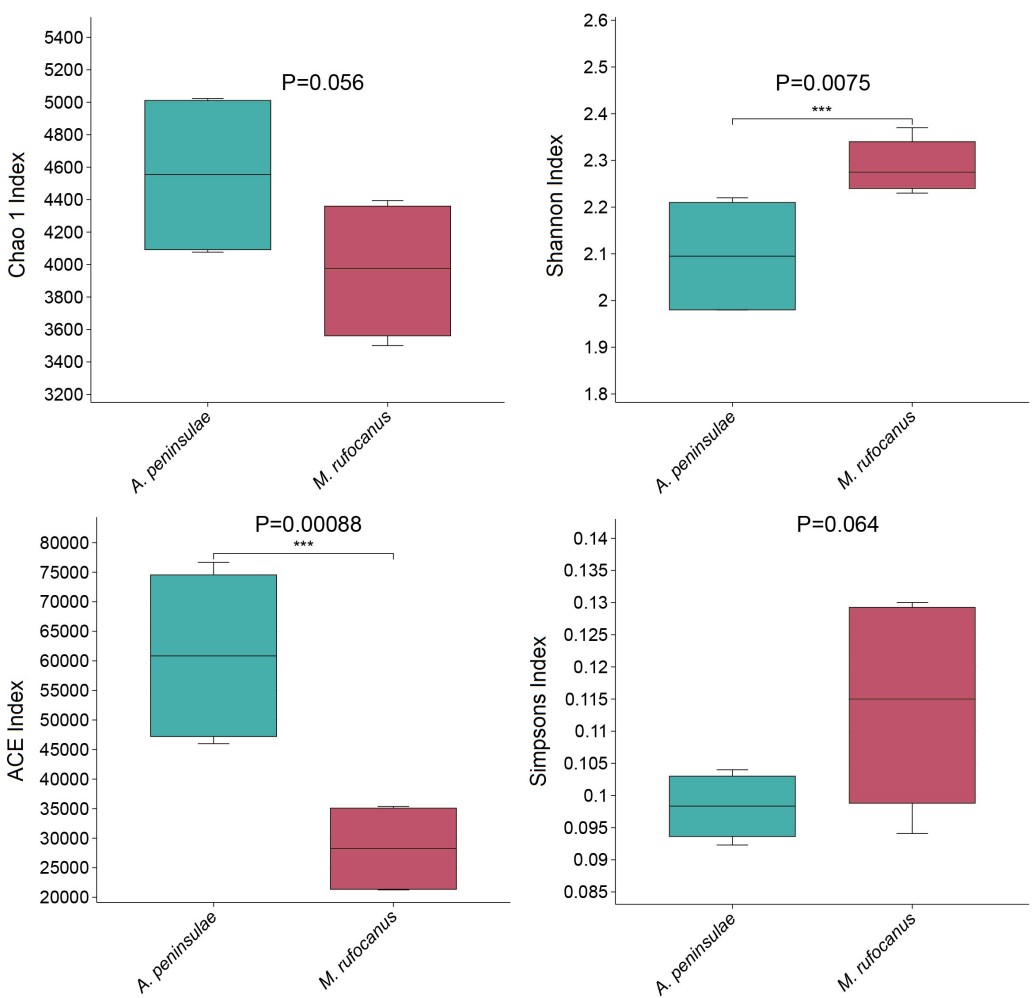

**Figure 3** **Diversity and richness indices.** Boxplots of the Chao1 (*t*-test), Shannon, ACE, and Simpson indices.

information processing, signaling, and cellular processes, as well as metabolic pathways involving carbohydrates, amino acids, cofactors, vitamins, energy, nucleotides, and lipids. Translation, replication and repair, and membrane transport were also identified as significant pathways (Fig. 6B).

The CAZy database is a knowledge-based resource specialized in the enzymes involved in synthesizing and degrading complex carbohydrates and glycoconjugates. By searching the CAZy database, unique genes corresponding to six CAZy modules and 100 CAZy families were identified within the gut microbiomes of *A. peninsulae* and *M. rufocanus* (Fig. S3). At the CAZy classification level A, glycoside hydrolases (GH), glycosyl transferases (GT), and carbohydrate-binding modules (CBM) were identified as the top three dominant enzyme families, constituting 93.28% to 97.41% of the enzyme families in the two species (Fig. S3A). At level B, the top 10 dominant enzyme families included GH3, GH31, GT2, GH13, CBM48, GT4, GH5, GT51, GH43, and GH29 (Fig. S3B).
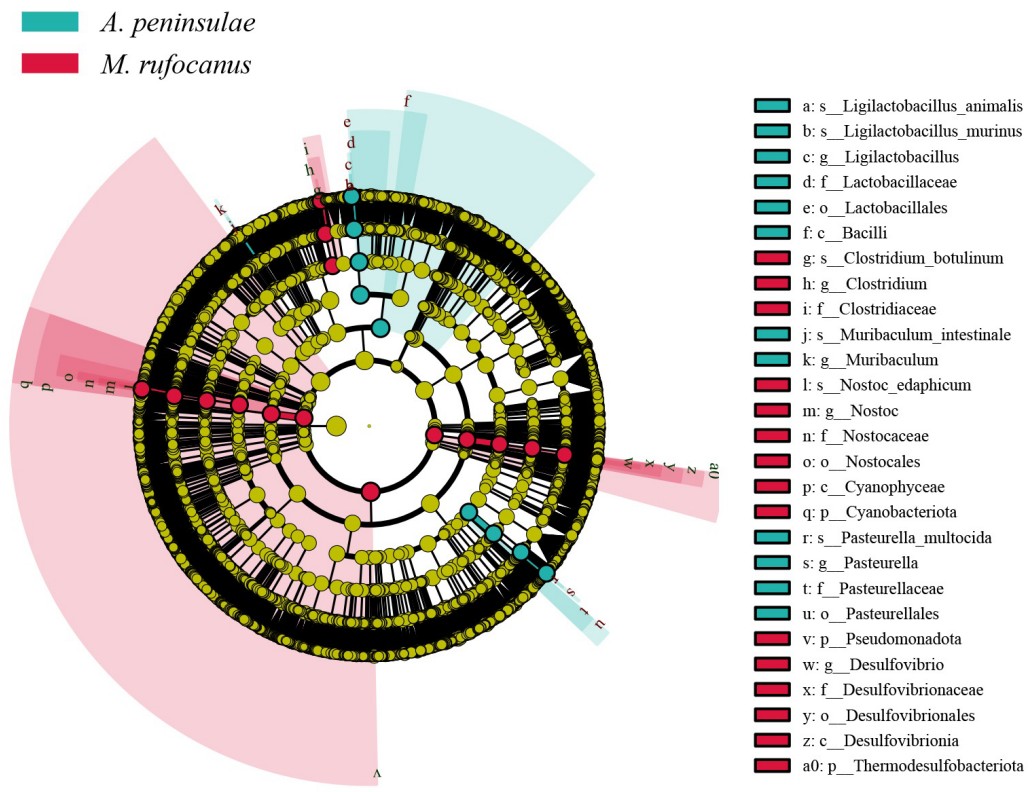

**Figure 4** Cladogram of the significantly different biomarkers from *A. peninsulae* and *M. rufocanus* guts (LDA > 4, *p* < 0.05).

The Bray–Curtis distances were calculated based on the genes within the KEGG and CAZy databases, followed by the generation of a PCA plot. Results indicated a distinct separation in the functional composition of the gut microbiomes between *A. peninsulae* and *M. rufocanus* (Figs. 7A and 7B). STAMP analysis was conducted to identify features with significant intergroup differences based on functional abundance (Fig. 7C). At KEGG level B, significant differences between *A. peninsulae* and *M. rufocanus* were observed in functions related to infectious diseases (viral and bacterial), transport and catabolism, the endocrine system, and cell growth and death. The CAZy database annotated six enzyme families—GH19, PL12, GH4, GH43, GH32, and GT1—that were significantly enriched in *A. peninsulae* (*p* < 0.05; Fig. S4). *M. rufocanus* also had a greater enrichment of enzyme families, specifically CE10, GH5, GH1, and GT43, compared to *A. peninsulae* (*p* < 0.05).

## DISCUSSION

*M. rufocanus* and *A. peninsulae*, common forest rodents, are known to cause rodent damage. They both maintain herbivorous diet. However, the mechanisms by which *M. rufocanus* and *A. peninsulae* adapt to their fibrous diet remain unclear. In this study, the composition and functional roles of gut bacteria in the two rodent species were investigated through metagenomic sequencing. The taxonomic assignment of genomic sequences revealed that

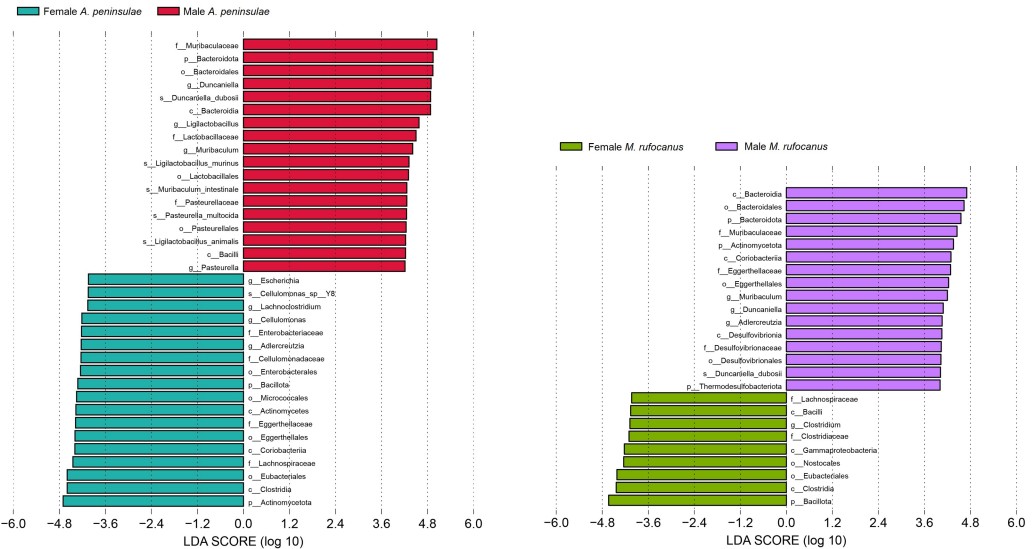

**Figure 5** Bar chart of the significantly different biomarkers from different sexes of *A. peninsulae* and *M. rufocanus* guts (LDA > 4, *p* < 0.05).

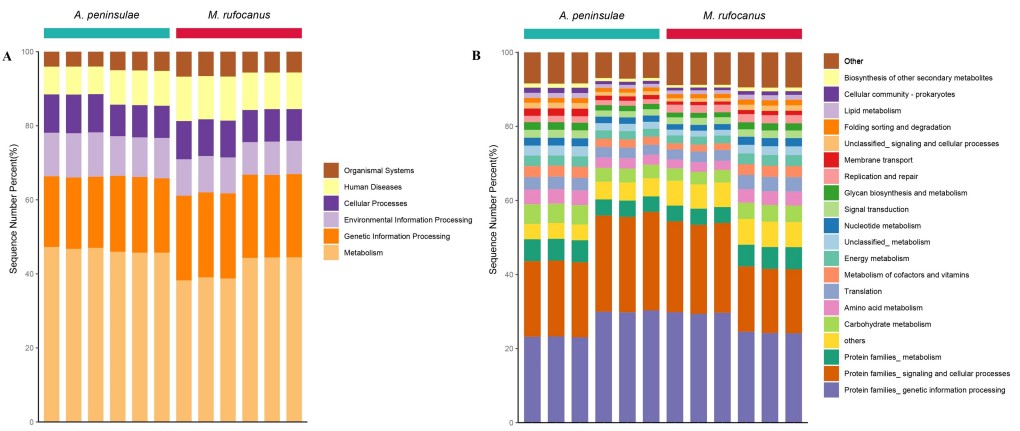

**Figure 6** Gut microbiome function of *A. peninsulae* and *M. rufocanus* at KEGG classification levels (A) and (B).

the gut microbiota of *M. rufocanus* and *A. peninsulae* encompassed 43 bacterial phyla. On average, Firmicutes, Bacteroidetes, Proteobacteria, and Actinobacteria constitute >89% of the total gut microbiota, which is typical in wild house mice (*Linnenbrink et al., 2013*; *Suzuki & Nachman, 2016*; *Wang et al., 2014*; *Weldon et al., 2015*). The higher diversity of these taxa suggests adaptations to break down fiber-rich, herbivorous diets. The dominance of Firmicutes may imply a role in carbohydrate fermentation, by potentially contributing to energy harvest, as was reported in laboratory mice (*Flint et al., 2012*; *Ley et al., 2008*). Additionally, Firmicutes produce essential vitamins, including vitamin K and biotin, which may not be readily available in the rodent diet (*LeBlanc et al., 2017*). Bacteroidetes

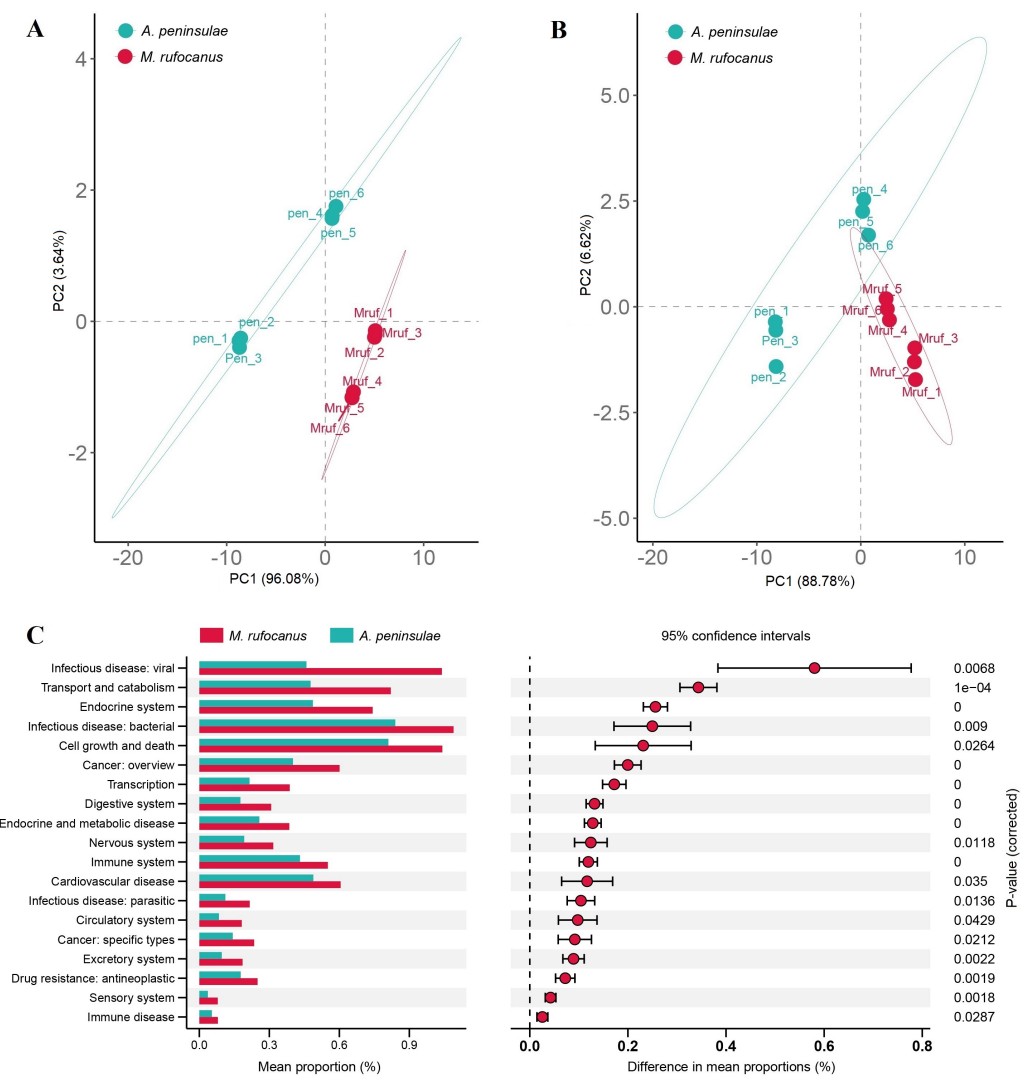

**Figure 7  Gut microbial diversity analysis between  *A. peninsulae* and *M. rufocanus*.** (A) PCA plot of the gut microbiome composition on KEGG level B annotation. (B) PCA plot of the gut microbiome composition on CAZy annotation. (C) Inferred differential functions based on KEGG annotation.

may also aid in the host's degradation of succinate, a key precursor for the synthesis of molecules such as gluconeogenic substrates and neurotransmitters (*Nuriel-Ohayon, Neuman & Koren, 2016*; *Waite & Taylor, 2014*). Proteobacteria contain enzymes capable of breaking down proteins, thereby supporting the host's growth and development (*Rawls et al., 2006*; *Shin, Whon & Bae, 2015*). Actinobacteria possess enzymes that can degrade complex and recalcitrant compounds, such as lignin and chitin, which are components of the rodent diet (*Bibb, 2005*). Apart from providing essential nutrients and energy through the fermentation of dietary fiber, the gut bacteria in rodents may also regulate immune function and protect against pathogens. Firmicutes modulate immune responses, thereby contributing to maintaining intestinal homeostasis and preventing inflammation (*Atarashi*

*et al., 2011*; *Furusawa et al., 2013*). Proteobacteria are also engaged in immune surveillance, playing a role in the recognition and response to pathogens (*Darbandi et al., 2022*). Certain Actinobacteria species have been identified as possessing anti-inflammatory properties, which aid in suppressing gut inflammation (*Watve et al., 2001*). A prior study has shown that an elevated ratio of Firmicutes to Bacteroidetes in the gut microbiota correlates with a higher dietary energy harvest (*Turnbaugh et al., 2006*). Results indicated only a minor difference in the Firmicutes to Bacteroidetes ratio between *M. rufocanus* (0.81) and *A. peninsulae* (1.01). One potential explanation for this similarity is that *M. rufocanus* and *A. peninsulae* occupy the same ecological niche and exhibit similar ecological traits within the study area. The two rodent species also share comparable feeding habits. Higher Simpson and Shannon indices in *M. rufocanus* suggest that the abundance and evenness of bacterial species in its gut are greater than those observed in *A. peninsulae*. Furthermore, STAMP analysis revealed that the gut Proteobacteria abundance in *M. rufocanus* was significantly higher than *A. peninsulae* ($P = 0.0062$). It is widely considered that species identity exerts a more substantial influence than environmental factors in shaping the gut microbiota of wild rodents (*Knowles, Eccles & Baltrūnaite, 2019*).

The common microbiota at the family level between *A. peninsulae* and *M. rufocanus* include Muribaculaceae, Lachnospiraceae, and Eggerthellaceae. Species in the Lachnospiraceae and Muribaculaceae families were abundant and specific to Murinae (rats and mice) (*Bowerman et al., 2021*; *Weinstein et al., 2021*). Enrichment of these bacterial families associated with the digestion of fibrous diets and detoxification of plant secondary compounds was a key functional adaptation in *A. peninsulae* and *M. rufocanus*. The family Muribaculaceae produces propionate, a fermentation end-product that aids in immune system regulation and possesses anti-inflammatory properties. Consequently, this bacterial family is linked to improved gut health and extended longevity in mice. Furthermore, species within this family exhibit varied responses to dietary interventions, such as acarbose treatment, indicating a role in starch fermentation and adaptability to dietary alterations (*Smith, Miller & Schmidt, 2021*; *Smith et al., 2019*; *Wang et al., 2023*). The Lactobacillaceae family comprises probiotic bacteria that inhibit the proliferation of harmful bacteria, enhance the gut-barrier function, and modulate the immune system. They also contribute to vitamin and SCFA production, essential for gut health maintenance, energy provision to the host, and immune system regulation (*Walter & O'Toole, 2023*). Members of the Lachnospiraceae family are prominent SCFA producers. They also participate in the fermentation of dietary fibers (*Vacca et al., 2020*). The family Eggerthellaceae, which encompasses the genus *Eggerthella*, is recognized for its role in the metabolism of dietary components and is linked to the production of health-promoting metabolites.

At the genus level, the common gut bacteria dominant in both rodent species were *Duncaniella*, *Adlercreutzia*, *Bacteroides*, *Streptomyces*, and *Clostridium*. *Duncaniella* was associated with disease variability in a mouse model of colitis. Specifically, *Duncaniella muricolitica* was identified as playing a dominant role in the dextran sulfate sodium mouse model of inflammatory bowel disease (*Chang et al., 2021*). The genus *Adlercreutzia* plays a role in the metabolism of dietary components, potentially influencing the host's adaptation and health (*Lu et al., 2021*). *Bacteroides* is one of the most abundant genera

in the gut microbiome and is renowned for its role in the breakdown of complex carbohydrates. Variations in the abundance of Bacteroides species have been linked to various health outcomes, including inflammation and gut-barrier integrity (*Lu et al., 2021*). *Streptomyces* produces a wide range of bioactive compounds, including antibiotics. Therefore, *Streptomyces* may contribute to the host's resistance to pathogens (*Bowerman et al., 2021*). Certain *Clostridium* clusters are less prevalent in laboratory mice than in wild-type mice, suggesting a role in host fitness and disease resistance (*Chang et al., 2021*). Therefore, these bacterial genera play a role in various aspects of the rodents' digestive and immune systems and energy metabolism.

The gut microbiomes of *M. rufocanus* and *A. peninsulae* exhibit distinct microbial compositions that reflect adaptations to their respective ecological niches. Both species are dominated by *D. duboisii*, suggesting its critical role in carbohydrate fermentation, similar to *Bacteroides* spp. (*Flint et al., 2012*). The presence of *M. intestinale* in both rodents underscores its importance in digesting complex carbohydrates, consistent with findings in other rodent studies (*Lagkouvardos et al., 2019*). Interestingly, *C. botulinum* is prominent in *M. rufocanus*, suggesting either a unique adaptation or potential health risk, whereas *P. multocida*, present in both species, may function as a stable commensal or opportunistic bacterium (*Wilson & Ho, 2013*). The presence of *A. equolifaciens* and *Adlercreutzia* sp. 8CFCBH1 suggests a diet rich in plant materials, as these bacteria are known for metabolizing isoflavonoids (*Clavel et al., 2006*). Additionally, *E. coli*, a common gut inhabitant, reflects host health and environmental factors (*Tenaillon et al., 2010*), whereas the presence of *F. plautii* suggests a role in metabolizing plant compounds. Notably, *M. rufocanus* hosts *D. fairfieldensis* and *B. viscericola*, whereas *A. peninsulae* includes *Lachnoclostridium* sp. YL32 and *Cellulomonas* sp. Y8, indicating dietary and environmental variations, with *Desulfovibrio* involved in sulfur metabolism and *Barnesiella* in pathogen resistance (*Buffie et al., 2015*).

Preliminary observations of sex-based differences suggest potential sex-specific adaptations, though further studies with balanced sample sizes are needed to confirm these trends. In male *A. peninsulae*, the presence of *D. duboisii, L. murinus, M. intestinale, P. multocida*, and *L. animalis* indicates a microbiome potentially oriented toward enhanced carbohydrate fermentation and pathogen resistance. *D. duboisii* and *M. intestinale* are recognized for their roles in carbohydrate metabolism, potentially providing dietary advantages (*Flint et al., 2012*; *Lagkouvardos et al., 2019*). *Ligilactobacillus* species, including *L. murinus* and *L. animalis*, are frequently associated with gut health and immunomodulation, potentially offering protective benefits against gastrointestinal disturbances (*Walter, 2008*). Conversely, the microbiome of female *A. peninsulae* is dominated by *Cellulomonas sp. Y8*, suggesting potential specialization in cellulose degradation, possibly linked to dietary variations or ecological niches favoring plant-based diets (*Stackebrandt, Rainey & Ward-Rainey, 1997*). This highlights the influence of dietary and environmental factors on sex-specific adaptations in the gut microbiome. In *M. rufocanus*, males predominantly harbor *D. duboisii*, indicating a similar emphasis on carbohydrate metabolism as observed in *A. peninsulae* males. Conversely, the microbiome of female *M. rufocanus* is characterized by a significant presence of *Clostridium* species.
*Clostridium*, recognized for its diverse metabolic capabilities, including fermentation and butyrate production, may contribute to energy harvesting and gut health maintenance (*Louis & Flint, 2009*).

The gut microbiota primarily functions in breaking down food and enhancing nutrient absorption for the host. Beneficial bacteria within the gut microbiota significantly contribute to the increased intake of proteins, sugars, and vitamins, thereby improving dietary component utilization. This study elucidated the functions of the gut microbiota in *A. peninsulae* and *M. rufocanus*, with gut bacteria in these rodent species enriched in various metabolic activities, including carbohydrate and amino acid metabolism. A plausible explanation is that host-derived carbohydrates and proteins are the primary nutrients influencing the composition of the resident bacteria (*Gibson & Roberfroid, 1995*). Numerous genes encoding carbohydrate-digestive enzymes, such as GHs, GTs, and CBMs, were identified within the gut bacteria of rodents, suggesting that these bacteria may compensate for the inability of the rodents to efficiently digest polysaccharides. Consequently, the metabolic potential of gut bacteria may be associated with the host's diet, as inferred from the enrichment of carbohydrate-active enzymes linked to plant polysaccharide degradation.

Functional analysis results revealed that gut microbial functions vary with host phylogeny. Based on STAMP analysis utilizing the CAZy database, significant differences were observed in the relative proportions of GH families between *M. rufocanus* and *A. peninsulae*. Notably, despite the distinct taxonomic compositions of the gut microbiota between *M. rufocanus* and *A. peninsulae*, their functional profiles exhibited high similarity. This pattern may reflect a convergent evolution of microbial communities, wherein phylogenetically divergent bacterial species have adapted to perform analogous metabolic functions that benefit their hosts (*Muegge et al., 2011*). Such functional convergence could be driven by host-specific selective pressures, such as dietary constraints or shared ecological niches (*Lozupone et al., 2012*). For instance, while GH family abundances varied between species (*e.g.*, GH19 in *A. peninsulae vs.* GH5 in *M. rufocanus*), both taxa maintained a robust capacity for carbohydrate metabolism, aligning with their herbivorous diets. The relative abundance of GH19, GH4, GH43, and GH32 was elevated in *A. peninsulae*, whereas GH5 and GH1 were more prevalent in *M. rufocanus*. The presence and relative abundance of specific GH families can vary markedly, indicative of the diverse dietary habits and ecological niches these animals inhabit. GHs play a crucial role in breaking down of complex carbohydrates (*Lee et al., 2014*) and are indispensable in the processing of various exogenous and endogenous glycoconjugates within the human gut microbiota (*Pellock et al., 2018*). The enhanced relative abundance of GH19, GH4, GH43, and GH32 in *A. peninsulae* could reflect an adaptive potential for processing complex carbohydrates, consistent with their herbivorous diet (*Batsaikhan et al., 2016*; *De Filippo et al., 2010*). These GH families are recognized for targeting a broad spectrum of substrates, including plant cell wall polysaccharides abundant in the herbivore and omnivore diets (*Bourne & Henrissat, 2001*). In contrast, the increased prevalence of GH5 and GH1 in *M. rufocanus* indicates a distinct metabolic specialization within this host's gut microbiota. GH5 and GH1 enzymes are known for their involvement in the degradation of cellulose and chitin,

respectively (*Cantarel et al., 2009*). This suggests that *M. rufocanus*, like *A. peninsulae*, possesses a gut microbiota adept at processing a diet rich in plant materials, with a particular focus on utilizing chitin from the exoskeletons of insects and fungi. However, competition for resource utilization exists between the two species. This competition can indirectly affect the survival, reproduction, and growth of the competing species due to the reduction of total resources, and interspecies interference arises from the utilization of shared resources. Niche differentiation in habitat, feeding habits, activity patterns, or other ecological characteristics is inevitable among them.

## CONCLUSION

In this study, we characterized the gut microbiome of two sympatric rodent species, *M. rufocanus* and *A. peninsulae*, through metagenomic sequencing. Our results indicated that the dominant phyla within the intestinal flora of both species were Firmicutes, Bacteroidetes, and Proteobacteria. The intestinal flora of *M. rufocanus* exhibited greater diversity than that of *A. peninsulae*. The functional profiles of the gut microbiota are predominantly associated with metabolism, genetic information processing, and environmental information processing. Notably, the metabolic capabilities of the gut bacteria, especially in terms of carbohydrate and amino acid processing, are closely aligned with the herbivorous diet of the host.

### Funding

This work was supported by Natural Science Foundation of Shandong Province (grant number: ZR2023MC016) and Key Research and Development and Promotion Project (Scientific and Technological Project) of Henan Province (grant number: 232102320299). The funders had no role in study design, data collection and analysis, decision to publish, or preparation of the manuscript.

### Grant Disclosures

The following grant information was disclosed by the authors:
Natural Science Foundation of Shandong Province: ZR2023MC016.
Key Research and Development and Promotion Project: 232102320299.

### Competing Interests

The authors declare there are no competing interests.

### Author Contributions

- Jing Cao conceived and designed the experiments, performed the experiments, analyzed the data, prepared figures and/or tables, authored or reviewed drafts of the article, and approved the final draft.
- Shengze Wang performed the experiments, analyzed the data, prepared figures and/or tables, and approved the final draft.

- Ruobing Ding performed the experiments, analyzed the data, prepared figures and/or tables, and approved the final draft.
- Yijia Liu performed the experiments, analyzed the data, prepared figures and/or tables, and approved the final draft.
- Baodong Yuan conceived and designed the experiments, analyzed the data, prepared figures and/or tables, authored or reviewed drafts of the article, and approved the final draft.

## Animal Ethics

The following information was supplied relating to ethical approvals (i.e., approving body and any reference numbers):

Laboratory Animal Ethics Committee of Shangqiu Normal University approval for this research (2022102).

## Data Availability

The datasets generated for this study are available at Sequence Read Archive (SRA) of NCBI: SRR28435458–SRR28435469.

## Supplemental Information

Supplemental information for this article can be found online at http://dx.doi.org/10.7717/peerj.19260#supplemental-information.

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
