# Peer review of "Comparative analyses of the gut microbiome of two sympatric rodent species, *Myodes rufocanus* and *Apodemus peninsulae*, in northeast China based on metagenome sequencing"

_PeerJ, doi:10.7717/peerj.19260_

## Round 0.1 · original submission · Major Revisions

Dear Dr. Cao, As you can see, all three reviewers liked the topic. However, all of them raised concerns regarding the structure, part of the methodology, organization or presentation of the data. Please address all these issues carefully while revising your manuscript.

Best regards,
Elisabeth Grohmann

Reviewer 1 ·

Basic reporting

The overall clarity of the manuscript is lacking. The work is repetitive, unnecessarily wordy and the lines of arguments are not fully developed. The structure of both intro and discussion does not follow a logical flow of arguments.

The literature references is ok but either relies on reviews instead of citing primary literature or uses medical or wildlife literature interchangeably and not always at the right spot. Important references for metagenomic work in the wildlife sector are missing. For example, Cabral et al, which looked at the metagenomic diversity of the herbivorous rodent Capybaras, should have been cited. Equally, work from the Kevin Kohl Lab and Simone Sommer Lab works on rodent microbiomes and is largely missing.

No hypothesis is presented.

Experimental design

Research question is not defined and not embedded in the primary literature.

The methodology described is appropriate but the investigation could have gone into more depth albeit limited by the 12 metagenomes. Apparent differences between the sexes are not explored.

Description of sequencing protocol was poor.

Validity of the findings

While the study is not novel in its approach it could redeem itself if it was written well and discussed their finding as case study and further evidence to theories and thoughts that are being developed in the field.

The discussion contains a whole paragraph that is thematically the same but with slightly different word choices. I want to believe this is a mistake rather than a copy/paste error from a language translation website or worse ChatGPT.

Additional comments

I do not wish to outright reject the article even though there would be enough grounds to do so, but allow the authors to address some of the main concerns.

These are:
1. Poor developed rationale and overall poor reporting/writing style
2. poor coverage of literature
3. Lack of an in-depth analysis of their findings.

·

Basic reporting

The paper is well-written, comes on time, and is easy to understand.

Experimental design

Thank you for having me on this manuscript revision. The paper by Cao and colleagues aims at identifying differences in the intestinal microbiomes of two species of rodents from North China, Apodemus peninsula and Myoides rufocanos. The manuscript is well written, and comes on time. I appreciate the topic in general, i.e. attempting a better understanding of the animal microbiomes beyond the human microbiome using shotgun technologies.
Major concerns:
I have one major concern. The author used MEGAN. As far as I understand, MEGAN cannot profile species that lack a representative on NCBI. Recent papers (https://doi.org/10.1016/j.celrep.2023.112464, https://doi.org/10.1038/s41587-023-01688-w) highlighted how assembly-derived species-level genome bins (SGB) allow the accurate profiling of mammalian systems different from man. The rodent microbiome is highly unexplored from the point of view of isolation and cultivation of microorganisms, and assuming a profile with isolates-based marker-based methods is not recommended. Therefore I doubt the conclusions obtained on the basis of the adopted tools are strong.

Validity of the findings

Related to these first concerns, the authors list an impressive number of all-levels taxa found in the samples, including a total of 6,406 species found in twelve samples. This number is impressive, especially considering the concern above.

Additional comments

Personally, I found an additional caveat of this manuscript that, potentially following from the previous concerns, focus is on families and genera while it could be prerogative of shotgun metagenomics allowing a greater focus on species and strains.

Minor concerns:
In the Discussion, italics of Genera are missing.

Reviewer 3 ·

Basic reporting

The authors provided sufficient background and a very thorough Discussion section!

I had three minor comments regarding the organization of the Introduction section:
L96: I suggest starting a new paragraph when authors discuss the impact of evolutionary history, or making the transition from diet effects to evolutionary history more seamless.

L103: Another request to improve the transition from changing environmental conditions / seasonal variations to the Sun. et al. study. Make it clear we are now focusing on bacteria (Clostridiales, Bacteroidales, etc) and not seasonal variation.

L111-112: I suggest opening this paragraph by explicitly stating the gap in knowledge and rationale for the study. What is missing, what is the puzzle or dilemma? Why did the authors study two sympatric rodent species?

Experimental design

The sample collection of the mice seemed straightforward and the authors employed laboratory and bioinformatics methods that were of high technical standard.

A few comments to encourage authors to include methodological details that are missing:

L153 How were traps baited? At what time of the day were they checked?
L155-156 Do you know the mice's age or sex?
L172 What software was used for the Trimming?
L176 Did you use default settings with MEGAHIT?
L176 Can you run QUAST and provide results on the quality of the assembly here?
L178 Did authors consider removing host DNA by mapping metagenomic reads to the host genomes?
L189 What taxonomic reference database does MEGAN use?

Validity of the findings

The authors' Results section was comprehensive and easy to follow! Findings were reported for both taxonomic data and functional data.

Additional comments

L55-57: I suggest authors highlight the predominant bacterial Families or genera here instead of Phyla, which are not as informative.

L57-59: This sentence can be deleted. It is too broad to be useful and the authors discuss functional profiles in L60-64.

L79: change to "abundance of probiotic species"
L106: add “species” before digest cellulose
L146: can remove sentence about analgesia since this is repeated again in L149-151
L164: Not sure why the font is much smaller in this section
L282: change to "based on trimmed metagenomic data"

L283-286: I do not see much value in reporting findings at KEGG Level A. They are too broad to be informative.

Figure 1: Can the authors place Fig 1A on top of Fig 1B? So that the plot is two rows instead of the current 1 row. This will make this plot easier to see!

Figure 3: Capitalize y-axis capitalize y-axis so it reads as "Chao 1 Index", "Shannon Index", "ACE Index" and "Simpsons Index"

---

## Round 0.2 · Minor Revisions

Dear Dr. Cao,

As you can see from the comments of the two reviewers, they are mostly impressed by the revisions you have done to the manuscript. But there are still some issues which should be fixed before the manuscript can be considered for publication in the journal.

Please deal with these issues carefully, as suggested by Reviewer 1.

Kind regards,
Elisabeth Grohmann

Reviewer 1 ·

Basic reporting

The manuscript is much improved.

I still think the writing isnt as clear, detailed and informative as I would like it, but it is sufficient for an exploration on such a rather descriptive study with low sample sizes. As an example of what could be improved. In the abstract you write: "The gut microbiota of A. peninsulae and M.
rufocanus showed unique functional profiles, particularly in the breakdown of complex
carbohydrates, reflecting their dietary habits and ecological niches." This could be more precise if the authors chose to actually dive into the dietary habits and ecological niches, rather than leaving the reader wondering what they are. e.g., "The gut microbiota of A. peninsulae and M.
rufocanus differed marginally in functional profiles, particularly in the breakdown of complex carbohydrates, which might reflect their distinct food preferences albeit both being herbivores with a substantial dietary overlap"

I would recommend though that the authors tone down the praises of their own work. For instance, rather than stating "This research contributed uniquely to the field by providing
novel insights into the gut microbiota of two understudied rodent species..." I think this overstates the findings and could be phrased more like "This research aligns with information on rodent gut microbiota in general but puts a spotlight on two understudied rodent species..."

Delete L114-118 from this paragraph because meerkats are not rodents.

Experimental design

The authors claim that there is no a priori assumption and hence no need to formulate a hypothesis. This is fine if the authors then declare their work as purely descriptive. However, I do think that given the large niche and diet overlap, one could expect that in spite of being two evolutionarily separate species there is substantial overlap in taxonomy and function of their gut microbial community.

Given the limited sample size, conclusions like thsi are simply overdrawn: "This study contributed to the understanding of the adaptive mechanisms of rodents coexisting within complex forest ecosystems. Furthermore, it established a microbial data foundation for the prevention and management of harmful microorganisms in the forestry sector" I suggest writing something more on the line of: "This study provides a baseline for understanding the gut microbial community and potential functions for two rodents coexisting in Chinese forests." Leave out the complex forest ecosystem because you havent tested or looked at different sites within the forest matrix. There could be differences (see Schwensow et al. 2022), but as far as I can tell from the text this was not tested and the sample size would be too small for this anyway. Leave out the prevention stuff. What would you even prevent? Its to risky to base claims about pathogenic bacteria on 6 data points per species. That would be ethically dubious.

Validity of the findings

This is fine. The authors have done a good job incorporating feedback.

As I said, at times, results are overstated but a slight toning down would go a long way.

I think its interesting that the functional profile is more similar than the taxonomic profile. This argues for convergence among distinct bacterial strains that have adapted to benefit their host species.

Reviewer 3 ·

Basic reporting

The authors carried out extensive revisions on their manuscript and this has resulted in a significantly improved manuscript with clear writing, specific objectives, and detailed Methods, Results, and Discussion sections. I think the manuscript is great!

Experimental design

The authors employed field, laboratory, and bioinformatics methods that were of high technical standard. The research questions are well defined.

Validity of the findings

The authors present their findings clearly, which are supported by the underlying data.

Additional comments

I think the authors revisions are excellent and the manuscript would be a great addition to the published microbiome literature.

---

## Round 0.3 · accepted · Accept

Thank you for the resubmission of a highly improved version of the manuscript, which I have assessed myself. In my opinion, the manuscript is now ready for publication.